# Dynamics and Pretransitional Effects in C_60_ Fullerene Nanoparticles and Liquid Crystalline Dodecylcyanobiphenyl (12CB) Hybrid System

**DOI:** 10.3390/nano10122343

**Published:** 2020-11-26

**Authors:** Sylwester J. Rzoska, Szymon Starzonek, Joanna Łoś, Aleksandra Drozd-Rzoska, Samo Kralj

**Affiliations:** 1Institute of High Pressure Physics Polish Academy of Sciences, Sokołowska 29/37, 01-142 Warsaw, Poland; starzonek@unipress.waw.pl (S.S.); joanna.los@unipress.waw.pl (J.Ł.); ola.drozdrzoska@gmail.com (A.D.-R.); 2Laboratory of Physics of Complex Systems, Faculty of Natural Sciences and Mathematics, University of Maribor, Koroška 160, 2000 Maribor, Slovenia; Samo.Kralj@um.si; 3Jožef Stefan Institute, Jamova 39, 1000 Ljubljana, Slovenia

**Keywords:** liquid crystal nanoparticles, fullerenes, dielectric spectroscopy, phase transitions, dynamics

## Abstract

The report shows the strong impact of fullerene C_60_ nanoparticles on phase transitions and complex dynamics of rod-like liquid crystal dodecylcyanobiphenyl (12CB), within the limit of small concentrations. Studies were carried out using broadband dielectric spectroscopy (BDS) via the analysis of temperature dependences of the dielectric constant, the maximum of the primary loss curve, and relaxation times. They revealed a strong impact of nanoparticles, leading to a ~20% change of dielectric constant even at *x* = 0.05% of C_60_ fullerene. The application of the derivative-based and distortion-sensitive analysis showed that pretransitional effects dominate in the isotropic liquid phase up to 65 K above the clearing temperature and in the whole Smectic A mesophase. The impact of nanoparticles on the pretransitional anomaly appearance is notable for the smectic–solid phase transition. The fragility-based analysis of relaxation times revealed the universal pattern of its temperature changes, associated with scaling via the “mixed” (“activated” and “critical”) relation. Phase behavior and dynamics of tested systems are discussed within the extended Landau–de Gennes–Ginzburg mesoscopic approach.

## 1. Introduction

Liquid crystals are soft materials where fluidity is matched to the limited crystalline order, leading to unique properties significantly influenced by continuous or weakly discontinuous phase transitions, associated with the emergence or disappearance of single elements of symmetry [1]. Nanoparticles are extremely small solids with diameters between 1 and 100 nm, directly bridging the macro- and atomic/molecular-scales and yielding properties that are essentially different from the bulk material [2]. Linking liquid crystals (LCs) and nanoparticles (NPs), one can obtain a unique soft matter system due to the beneficial combination of extraordinary features of both species, enhanced by qualities emerging due to the host–guest interactions. Unique properties of LCs + NPs composites can be tuned by changing the concentration, type, and topology of nanoparticles [2,3,4]. The cognitive and fundamental progress in describing such systems, particularly those related to distortions in symmetry introduced by nanoparticles (including topological defects), is essential for the development of innovative applications of LCs + NPs nanocolloids [3,4,5,6,7]. From the fundamental point of view, the relatively simple characteristics of such a “composite” system are essential for numerical and theoretical modeling [3,4,8,9,10,11,12,13,14]. The evidence for assembling nanoparticles, particularly in smectogenic LC matrix, which can create an “orientation field” responsible for some unique features, is also notable [15,16,17,18,19]. In the group of nanoparticles, fullerenes play an exceptional role as the dopant to LC matrixes due to their very small size matched with extraordinary features. These have caused a notable boost in studies of LC + fullerene composites in recent years [20,21,22,23,24,25,26,27,28,29,30,31,32,33,34,35,36,37]. Still emerging fundamental challenges are supported by advanced possibilities of applications in innovative electrooptical devices [3,4,16,27,33,37]. First, new features of LC-type materials can be achieved without new chemical synthesis. Second, they can be smoothly tuned by changing the concentration and characteristics of nanoparticles. Finally, emerging and future applications can benefit from “added value features absent in pure LC materials” [3,4].

For the *Physics of Liquid Crystals*, phase transitions, related pretransitional effects, and dynamics constitute the basic reference for validating theoretical models [1,38,39]. However, such fundamental insights for LCs + NPs composites are still at the very beginning [17,18,19]. The broadband dielectric spectroscopy (BDS) constitutes here the key experimental tool due to the possibilities of the advanced investigations of dynamic properties matched with the sensitivity to intermolecular interactions or dipole–dipole arrangements. The possibilities of the scan, covering even 15 decades in time/frequency within a single measurement, are unique [40].

Notwithstanding, none of the studies carried out have addressed fundamental issues regarding the characterization of phase transition and dynamics presented in the given report. The available experimental evidence for LC + NP composites is limited regarding both dynamics and pretransitional behavior, despite their significance for the development of theoretical modelings. For instance, the dynamics insights suggest only the Arrhenius (A) or Vogel–Fulcher–Tammann (VFT) Super-Arrhenius (SA) portrayal [3,4,15,16,20,22,23,24], although the general validity of such scaling in “pure” LC compounds has been clearly questioned [41,42,43,44,45]. Tests of the quasi-critical pretransitional behavior for static properties, so significant for “pure” LC compounds [46,47,48,49,50,51,52,53,54], is still at the beginning in nanocomposites. The authors of this report tested the evolution of the dielectric constant in LC + BaTiO_3_ nanoparticle composites (diameter *d*~50 nm) for two “classical” rod-like liquid crystalline compounds: pentylcyanobiphenyl (5CB) and dodecylcyanobiphenyl (12CB) [17,18,19]. They showed the preservation of the form of the critical-like behavior, matched with strong changes in the isotropic liquid–mesophase transitions’ discontinuities, as well as the value of the dielectric constant, even for tiny concentrations of nanoparticles. To the best of the author’s knowledge, there have been no such studies for other types of nanoparticles, particularly fullerenes.

This report presents the results of studies in smectogenic liquid crystal 12CB and its nanocolloids with tiny concentrations of C_60_ fullerene (d≈1 nm). Fullerenes, including C_60_, are unique nanoparticles: despite their low dielectric constant and dimension, they constitute a unique addition to LC host matrix, strongly changing properties. This can be linked to the ability of fullerenes to cause guest–host interactions [20,21,22,23,24,25,26,27,28,29,30,31,32,33,34,35,36,37]. The applications of the innovative distortion-sensitive analysis presented below revealed new features of the tested LCs–NPs composites, that have been hidden thus far. Notable is the exceptional, long-range impact on the pretransitional behavior and the “glassy” complex dynamics. The latter is explained by the enhanced Landau–de Gennes–Ginzburg mesoscopic model [14,38], taking into account hallmarks of topological defects introduced by nanoparticles.

## 2. Materials and Methods

### 2.1. Samples Preparation

Dodecylcyanobiphenyl (12CB) belongs to the “classical” homologous series of LC compounds: *n*-alkyl-cyanobiphenyls (*n*CB). Their geometry is approximately rod-like, with a relatively large permanent dipole moment (μ≈5D), parallel to the long molecular axis. For 12CB, the following mesomorphism occurs: *Isotropic Liquid* (I)→$T_{I-SmA}=331.3K$→*Smectic A (SmA)*→$T_{SmA-Solid}=302.4K$→*Solid* (S) [1]. The approximate length of 12CB molecule is ca. 2.3 nm and the width 0.6 nm. The anisotropy of dielectric permittivity Δε≈16, which is related to $\varepsilon_{\left|\right|}\approx 18.5$, and $\varepsilon_{⊥}\approx 2.5$ [1]. Fullerene C_60_ nanoparticles (diameter d≈1nm) were purchased from Sigma-Aldrich. From dielectric tests in dilute solution, the dielectric constant of such fullerenes was estimated as ε≈2.2 [25]. The 12CB and C_60_ nanoparticles were prepared in a dry box. LC samples were carefully degassed prior to measurement and subsequently sonicated for 1 h and *T* = 60 °C to avoid sedimentation. The latter was also the reason for reducing tests to 0.1%. Samples were placed in a flat-parallel gold-coated capacitor manufactured by Invar. The basic gap of the capacitor was *d* = 0.2 mm. The “check-tests” were also carried out for *d* = 0.02 mm, and after experiments, samples were always retested at room temperature to confirm the lack of sedimentation. Even for concentrations x~10%, no significant impacts of sedimentation effects were observed.

### 2.2. Experimental Procedures

Broadband dielectric spectroscopy (BDS) studies were carried out using the Novocontrol BDS impedance analyzer yielding 5–6 digits resolution, with a Quattro Novocontrol temperature control unit. The set-up directly yielded the real and imaginary parts of dielectric permittivity: $\varepsilon^{*}=\varepsilon^{\prime }+i\varepsilon^{\prime \prime }$, with 5–6 digits resolution and the temperature control ±0.01 K.

Dielectric spectra, detected as described in the given report studies, are illustrated in Figure 1. The dielectric constant (ε) was determined from the stationary, frequency-independent domain of its real part: for the results presented below, f=10 kHz was taken as the reference, i.e., $\varepsilon =\varepsilon^{\prime }\left(f=10\right)$ kHz [40]. The second part of the analysis was related to loss curves ($\varepsilon^{\prime \prime }\left(f\right)$), reflecting basic “dynamic” features. Relaxation times were determined using frequencies of loss curves’ peaks $\tau =1/2\pi f_{peak}$ [40]. Generally, loss curves’ peaks are described by two parameters $\left(f_{peak},\varepsilon_{peak}^{\prime \prime }\right)$. The evolutions of the height of the loss curve $\varepsilon_{peak}^{\prime \prime }\left(f,T\right)$, which can be linked to the maximal energy coupled to the given relaxation process/mode [8], were also tested.

### 2.3. Data Analysis and Modeling

This section presents a brief summary of the experimental results for the pretransitional effects and dynamics of rod-like LC materials, which are essential for the research presented below. The analysis of the temperature evolution of the dielectric constant in the isotropic phase revealed the following behavior [46,47,48,49,50,51,52,53,54]:(1)$$\varepsilon \left(T\right)=\varepsilon^{*}+a\left(T-T^{*}\right)+A{\left(T-T^{*}\right)}^{\phi }$$
where $T^{*}>T^{C}$, and $T^{*}=T^{C}-\Delta T^{*}$ is the extrapolated temperature of a hypothetical continuous phase transition; $T^{C}=T_{ISmA}$ is the clearing temperature of the Isotropic–Smectic A weakly discontinuous phase transition; $\Delta T^{*}$ is the measure of the discontinuity of the given phase transition. The power exponent ϕ=1−α=1/2; the exponent α=1/2 is related to the pretransitional anomaly of the heat capacity (specific heat) [54].

The portrayal via Equation (1) was introduced for the isotropic–nematic (*I–N*) transition [42,47,48,51,52,53,54] and later also for the isotropic–smectic A (*I*–Sm*A*) [49] and isotropic–Smectic E [50] transitions. Such general validity results from the fact that dielectric constant measurements detect the orientational arrangement of permanent dipole moments within pre-mesomorhic fluctuations [48,51,53,54]. This is caused by the equivalence of $\vec{n}$ and $-\vec{n}$ directors describing the dominated orientation of rod-like molecules and occurring for all mesophases recalled above [1]. For the permanent dipole moment parallel to the long molecular axis, this leads to the cancellation of the related contribution to the dielectric constant within premesomorphic fluctuations [51,52,53,54]. Consequently, the dielectric constant of fluctuations is less than for the surrounding isotropic liquid, i.e., $\varepsilon_{fluct.}^{meso.}<<\varepsilon_{Iso.}$. At some distance from the clearing temperature, dε/dT=0, which is associated with the crossover dε/dT>0 (close to $T^{C}$, the domination of the “antiparallel arrangement”)→dε/dT<0 (remote from $T^{C}$ the domination of the “parallel arrangement” with respect to the electric field) [54].

In ref. [42], for the isotropic phase of 5CB with the isotropic–nematic transition, it was shown that similar behavior is exhibited in the maximum of the primary loss curve, namely:(2)$$\varepsilon_{peak}^{\prime \prime }\left(T\right)=\varepsilon_{peak}^{*}+a_{peak}\left(T-T^{*}\right)+A_{peak}{\left(T-T^{*}\right)}^{\phi }$$

It is notable that Equation (1) is related to the same frequency for all temperatures, whereas in Equation (2) each tested temperature is associated with a different frequency, i.e., $\varepsilon_{peak}^{\prime \prime }\left(T\right)=\varepsilon_{peak}^{\prime \prime }\left(T,f\right)$. The link between the dielectric constant and the height of the loss curve is simple within the basic Debye model, associated with a single relaxation time, namely: $2\varepsilon_{peak}^{\prime \prime }=\varepsilon -\varepsilon_{\infty }$ [40], where $\varepsilon_{\infty }$ denotes dielectric permittivity for the “infinite” frequency, where the impact of permanent dipole moment is absent, and no pretransitional behavior can be expected from this contribution [40]. However, in the isotropic phase of liquid crystalline compounds, there is a distribution of the relaxation times, leading to the broadening of the loss curve on cooling towards the clearing temperature [42,54]. This should influence the height of the loss curve but seems to be neutral regarding the functional form of $\varepsilon_{peak}^{\prime \prime }\left(T\right)$ [42]. In ref. [17] it was shown that the pretransitional effect, described in a manner that is parallel to Equation (1), also appears in the Sm*A* mesophase:(3)$$\varepsilon \left(T\right)=\varepsilon^{**}+a^{SmA}\left(T^{**}-T\right)+A^{SmA}{\left(T^{**}-T\right)}^{\phi }$$
where $T^{**}<T^{C}$ and $T^{**}=T^{C}+\Delta T^{**}$ is the temperature of the hypothetical continuous phase transition detected when observed from the mesophase side of the clearing temperature. $\Delta T^{**}$ is the metric of the discontinuity of the *I*–Sm*A* phase transition, detected from the mesophase side.

It is worth noting that this “mesogenic” (i.e., in the Sm*A* phase) pretransitional effect should be linked to isotropic liquid fluctuations (“droplets”) within the liquid crystalline “matrix”, and then $\varepsilon_{fluct.}^{Iso.}>>\varepsilon_{meso..}$.

The peak of the dielectric loss curve is characterized by the pair $\left(\varepsilon_{peak}^{\prime \prime },f_{peak}\right)$. The second component estimates the relaxation time: $\tau =1/2\pi f_{peak}$ for the given relaxation process. In the isotropic liquid phase, it can be recognized as the primary (*alpha*) process [40]. In the mesophase, the LC symmetry causes the split into two modes (see below) [1]. The δ-mode in the Sm*A* phase can be considered as the continuation of the *alpha*-relaxation process that is dominant in the isotropic liquid phase. There is considerable evidence indicating the simple Arrhenius temperature behavior for relaxation times, both in the isotropic liquid and LC mesophases ([1,38,39], and refs therein). However, the increase in the tested temperature range yielded clear evidence for the Super-Arrhenius (SA) behavior [40], namely [41,42,43,44]:(4)$$\tau \left(T\right)=\tau_{\infty }\exp\left(\frac{E_{a}\left(T\right)}{RT}\right)$$
where $E_{a}\left(T\right)$ denotes the apparent activation energy, *R* denotes gas constant, and $\tau_{\infty }$ denotes the pre-factor.

The above relation simplifies “down” to the basic Arrhenius equation if in the given temperature domain $E_{a}\left(T\right)=E_{a}=const$, and then $E_{a}=d\ln\tau \left(T\right)/d\left(1/T\right)$; otherwise, for the temperature-dependent activation energy one obtains $E_{a}=d\ln\tau \left(T\right)/d\left(1/T\right)$, where $H_{a}\left(T\right)$ stands for the apparent activation enthalpy [44,45,55]. The SA equation cannot be applied directly for portraying experimental data due to the lack of the general form of the activation energy, which causes substitute equations to be required [40]. The most popular is the Vogel–Fulcher–Tammann (VFT) relation [40]:(5)$$\tau \left(T\right)=\tau_{\infty }\exp\left(\frac{B}{T-T_{0}}\right)=\tau_{\infty }\exp\left(\frac{D_{T}T_{0}}{T-T_{0}}\right)$$
where $T_{0}$ is the extrapolated VFT singular [8].

The comparison of Equations (1) and (2) yields the VFT approximation of the activation energy via: $E_{a}\left(T\right)=RTB/\left(T-T_{0}\right)=RD_{T}T_{0}/\left[\left(T-T_{0}\right)/T\right]$. The parameter $D_{T}$ refers to the fragility strength. It is linked to the fragility coefficient $m=m_{P}\left(T=T_{g}\right)=d\log_{10}\tau \left(T\right)/d\left(T_{g}/T\right)$, which is one of the basic metrics of glassy dynamics: $D_{T}\propto 1/m$. $T_{g}$ denotes here the experimentally detected or extrapolated glass temperature estimated via the condition $\tau \left(T_{g}\right)=100$ s.

The linearized distortions’ sensitive analysis [55] has shown that better fitting quality is obtained for the portrayals of critical-like behavior, namely [41,42,43,44]:(6)$$\tau \left(T\right)=\tau_{0}{\left(T-T_{C}\right)}^{-\phi }$$
the exponent ϕ=1.5−2.5 ($\tau \left(T_{C}\right)\sim 10^{-7}$ s) in the isotropic phase and ϕ=8−10 on cooling towards the “glass temperature” ($\tau \left(T_{C}\right)>>10^{2}$ s). Recently, the new equation linking the “activated” (Equation (5)) and “critical” (Equation (6)) behavior, for the complex dynamics in glassy and soft matter systems, has been proposed [45]:(7)$$\tau \left(T\right)=C{\left(\frac{T-T^{*}}{T}\right)}^{\Omega }\exp\left(\Omega \frac{T-T^{*}}{T}\right)$$
where the prefactor $C=\tau_{0}\tau_{\infty }$ and $T^{*}$ is the extrapolated singular temperature.

Equations (5)–(7) are associated with three fitted parameters, the number considered as the indicator for the optimal relation describing the non-Arrhenius dynamics [40]. Notably, Equation (7) was derived from the new “universal” form of the apparent fragility, namely [45]:(8)$$m_{P}\left(T\right)=\frac{d\log_{10}\tau \left(T\right)}{d\left(T_{g}/T\right)}=\frac{A}{T-T^{*}}$$
where $T_{g}$ denotes the extrapolated hypothetical glass temperature related to the condition:

$\tau \left(T_{g}\right)=100$ s, and $m_{P}\left(T\right)$ is known as the “apparent fragility” or “steepness index”.

One can also consider the apparent activation enthalpy $H_{a}\left(T\right)$ to obtain the parallel of Equation (8) [45]:(9)$$m_{P}\left(T\right)=\frac{d\log_{10}\tau \left(T\right)}{d\left(T_{g}/T\right)}=\frac{1}{T_{g}\ln10}\frac{\ln\tau \left(T\right)}{d\left(1/T\right)}=k\frac{\ln\tau \left(T\right)}{d\left(1/T\right)}=k\times H_{a}\left(T\right)$$

In the analysis of experimental data, the nonlinear, multiparameter fitting is most often used—this is the case regarding both dynamics and pretranstional effects [1]. Nevertheless, such a commonly used procedure has to yield notable uncertainties/fitting errors. This fact, often “softly” discussed in research papers, is significant even for the three most basic adjustable parameter fittings. To reduce this parasitic factor, one should analyze at least 10x more experimental data than the number of fitted parameters in the extended range of temperatures and use high-resolution experimental data. In this report, such practice is supported by the derivative-based and distortion-sensitive analysis, which reduced the number of fitting parameters and made it possible to obtain optimal values of parameters from the linear regression analysis, thus also yielding their real fitting error [44,45,55].

We model the phase behavior of mixtures using Landau–de Gennes–Ginzburg-type mesoscopic approach [5,14]. Of interest is the impact of spherical nanoparticles on the ordering of 12CB liquid crystal. In 12CB, the coupling between orientational and translational ordering is strong enough to exhibit direct 1st order isotropic to smectic A phase transition at the critical temperature $T_{C}=T_{I=SmA}$ when the temperature is reducing. In the following, we present a simple model that allows us to estimate pretransitional ($T>T_{C}$), critical $T\sim T_{C}$, and smectic structural ($T<T_{C}$) behavior. In our approximate analysis, we set the parameters such that nematic ordering is uniaxial, and we neglect biaxial states. The orientational and translational degrees of freedom are described in terms of the nematic tensor $\underline{Q}=S\left(\vec{n}⊗\vec{n}-\underline{I}/3\right)$ and the smectic complex $\psi =\eta e^{i\phi }$ order parameter, respectively. The nematic director field $\vec{n}$ points, along with the local average uniaxial orientation of LC molecules and the uniaxial order parameter S, measure the extent of fluctuations about $\vec{n}$. The smectic phase field ϕ locates the smectic layers, and the extent of layer order is described by the translational order parameter η. In the equilibrium Sm*A* phase, the smectic layers are stacked along the nematic director field (which is spatially uniform) with the layer spacing $d_{0}=2\pi /q_{0}$. This layer configuration is given by $\phi =q_{0}\vec{n}\cdot \vec{r}$. Of interest are mixtures of 12CB LC and NPs, where we assume that the concentration of spherical NPs is relatively low. We set the parameters such that NPs are essentially homogeneously distributed and exhibit short-range anchoring interaction at the NP–LC interface, which tends to align LC molecules along a local interface surface normal (the so-called homeotropic anchoring). With this in mind, we express the free energy of the system as the sum of volume (*f_V_*) and NP–LC interface (*f_i_*) free energy contributions $F=∭f_{v}d^{3}\vec{r}+N_{NP}\iint f_{i}d^{2}\vec{r}$, where *N_NP_* stands for the number of NPs. The volume contribution consists of the nematic condensation $\left(f_{c}^{\left(n\right)}\right)$, smectic condensation $\left(f_{c}^{\left(s\right)}\right)$, nematic elastic $\left(f_{e}^{\left(n\right)}\right)$, external electric field $\left(f_{f}\right)$, smectic elastic $\left(f_{e}^{\left(n\right)}\right)$, and coupling term $\left(f_{c}\right)$. We express these in terms of the order parameter as
(10)$$f_{c}^{\left(n\right)}=\frac{3}{2}a^{\left(n\right)}(T-T_{n}^{*})Tr{\underline{Q}}^{2}-\frac{9}{2}b^{\left(n\right)}Tr{\underline{Q}}^{3}+\frac{9}{4}c^{\left(n\right)}{\left(Tr{\underline{Q}}^{2}\right)}^{2},\;f_{c}^{\left(s\right)}=a^{\left(s\right)}(T-T_{s}^{*}){\left|\psi \right|}^{2}+b^{\left(s\right)}{\left|\psi \right|}^{4}+c^{\left(s\right)}{\left|\psi \right|}^{6},\;f_{c}=-\frac{3D_{c}q_{0}^{2}}{2}\nabla \psi^{*}\cdot \underline{Q}\nabla \psi .\;f_{f}=-\frac{3\varepsilon_{0}\Delta \varepsilon }{2}\vec{E}\cdot \underline{Q}\vec{E},\;f_{e}^{\left(n\right)}=\frac{L}{2}{\left|\nabla \underline{Q}\right|}^{2},\;f_{e}^{\left(s\right)}=C_{\left|\right|}{\left|\left(iq_{0}\vec{n}-\nabla \right)\psi \right|}^{2}+C_{⊥}{\left|\left(\vec{n}\times \nabla \right)\psi \right|}^{2}\;f_{i}=-\frac{3w}{2}\vec{v}\cdot \underline{Q}\vec{v}$$

The numerical coefficients are introduced for later convenience. The condensation terms and the coupling term determine the equilibrium value of the nematic and smectic order. The quantities $a^{\left(n\right)}$, $b^{\left(n\right)}$, $c^{\left(n\right)}$, $a^{\left(s\right)}$, $b^{\left(s\right)}$, $c^{\left(s\right)}$, *D_C_*, $T_{n}^{*}$, $T_{s}^{*}$ are positive material constants. The nematic elastic term is expressed in terms of a single positive bare elastic constant L. It roughly holds that $K\sim LS^{2}$ where *K* stands for the representative average Frank elastic constant. It tends to establish uniform nematic ordering along a single symmetry breaking direction. The quantity $\vec{E}$ stands for an external electric field, $\varepsilon_{0}$ is electrical permittivity, and Δε measures the electric field anisotropy. In LCs with positive anisotropy, uniaxial $\vec{n}$ tends to be aligned along $\vec{E}$. The smectic elastic term is weighted by the smectic compressibility ($C_{\left|\right|}$) and bending $\left(C_{⊥}\right)$ elastic constant. These positive constants enforce equidistant layer distance and alignment of the smectic layer normal along $\vec{n}$, respectively. Henceforth, we neglect an anisotropy in smectic elastic constants and set $C=C_{\left|\right|}=C_{⊥}$. The interface term is weighted with a positive anchoring constant, which favors alignment of $\vec{n}$ along local interface surface normal $\vec{v}$. Note that LC phase behavior strongly depends on the strength of the coupling constant *D_c_*. On increasing *D*_c_, we obtain three qualitatively different regimes, separated by values $D_{c}^{\left(1\right)}$ and $D_{c}^{\left(2\right)}$. On decreasing temperature starting from the isotropic phase, one encounters the following phase behavior. In the regime $D_{c}<D_{c}^{\left(1\right)}$, there is a phase sequence *I–N*–Sm*A*, where the *I–N* and *N*–Sm*A* transitions are discontinuous and continuous, respectively. Above the tricritical point $D_{c}=D_{c}^{\left(1\right)}$ and for $D_{c}<D_{c}^{\left(2\right)}$, the *N*–Sm*A* phase transition is discontinuous. Above $D_{c}^{\left(2\right)}$, we have a direct discontinuous *I*–Sm*A* phase transition. Regarding the impact of “small” nanoparticles on smectogenic LC, it is necessary to first introduce the volume concentration of NPs
(11)$$p=\frac{N_{NP}v_{NP}}{V}$$where *v_NP_* is the volume of a nanoparticle, and *V* is the volume of a system. In our case, NPs are spherical, characterized by radius *r*, therefore $v_{NP}=4\pi r^{3}/3$.

If NPs are homogeneously distributed within a sample, then their average separation is given by
(12)$$l_{NP}={\left(\frac{4\pi }{3p}\right)}^{1/3}r$$

For example, for *p* = 0.001 and *r* = 1 nm, Equation (12) yields $l_{NP}\sim 16\;\mathrm{nm}$. The volume concentration and mass concentration *c* are, in the diluted regime, related via $c\sim p\left(\rho_{NP}/\rho_{LC}\right)$, where $\rho_{NP}\sim 1.7\;g/{\mathrm{cm}}^{3}$ and $\rho_{LC}\sim 1\;g/{\mathrm{cm}}^{3}$ are mass densities of fullerene and LC, respectively. Several other material-dependent lengths play an important role in our treatment. These are the nematic/orientational order parameter length $\left(\xi_{n}\right)$, smectic order parameter length $\left(\xi_{s}\right)$, nematic penetration length (λ), external field coherence length $\left(\xi_{E}\right)$, and surface anchoring extrapolation length $\left(d_{e}\right)$. In terms of coefficients introduced in our model, we express these as
(13)$$\xi_{n}\sim \sqrt{\frac{L}{a^{\left(n\right)}\left(T-T_{n}^{*}\right)}},\;\xi_{n}\sim \sqrt{\frac{C}{a^{\left(s\right)}\left(T-T_{s}^{*}\right)}},\;\lambda \sim \sqrt{\frac{K}{Cq_{0}^{2}\eta^{2}}},\;\xi_{E}\sim \sqrt{\frac{K}{\varepsilon_{0}\Delta \varepsilon E^{2}}},\;d_{e}\sim \frac{K}{w}$$

Note that we expressed the order parameter correlation lengths $\xi_{n}$ and $\xi_{s}$ at temperatures above the relevant phase transition temperature. In our approximate treatment, we describe the average free density $\bar{f}\sim F/V$ as
(14)$$\bar{f}={\bar{f}}_{v}+\frac{3p}{r}{\bar{f}}_{i},$$where $\bar{\left(\ldots \right)}$ stands for the spatial average.

## 3. Results and Discussion

Figure 2 shows temperature evolutions of the dielectric constant and the maximum (peak) of loss curves for 12CB and 12CB + C_60_ fullerene nanocomposites in the Isotropic, Smectic A, and Solid phases. The behavior in the isotropic liquid phase can be well portrayed via Equations (1) and (2). The evolution in the Smectic mesophase follows Equation (3) for the dielectric constant, and can be described via the relation:(15)$$\varepsilon_{peak}^{\prime \prime }\left(T\right)=\varepsilon_{peak}^{\prime \prime }+a_{peak}^{SmA}\left(T^{**}-T\right)+A_{peak}^{SmA}{\left(T^{**}-T\right)}^{\phi }$$
where $T^{**}$ is for the extrapolated singular temperature.

The exponent ϕ=1−α≈0.5 for all tested LC-based systems—both in the isotropic liquid and Smectic A, in pure 12CB and its composites with C_60_ fullerene. However, significant influence on the phase transition discontinuity $\Delta T=T_{I-SmA}-T^{*}$ takes place.

The estimation of fitted parameters was supported by the derivative analysis of experimental data, shown in Figure 3 and Figure 4. They revealed the pronounced pretransitional anomalies for $T\to T_{I-SmA}$, well portrayed by relations resulting from Equations (1)–(3) and (15), namely:(16)$$\frac{d\varepsilon \left(T\right)}{dT}=A\phi {\left(T-T^{*}\right)}^{-\alpha }+a$$
(17)$$\frac{d\varepsilon_{peak}\left(T\right)}{dT}=A_{peak}\phi {\left(T-T^{*}\right)}^{-\alpha }+a_{peak}$$
(18)$$\frac{d\varepsilon \left(T\right)}{dT}=A^{SmA}\phi {\left(T^{**}-T\right)}^{-\alpha }+a^{SmA}$$
(19)$$\frac{d\varepsilon_{peak}\left(T\right)}{dT}=A_{peak}^{SmA}\phi {\left(T^{**}-T\right)}^{-\alpha }+a_{peak}^{SmA}$$

Solid curves in Figure 3 and Figure 4, based on Equations (16)–(19), clearly show that the impact of pretransitional fluctuations extend up to at least $T^{C}+65K$ in the isotropic liquid phase and covers almost the whole liquid crystalline mesophase. Small distortions appear only in the immediate vicinity of the Sm*A*–Solid-phase transition, particularly in nanocomposites. In the opinion of the authors, they indicate the presence of the SmA→S pretransitional effect, although too weak for a reliable parameterization. It is accompanied by a pretransitional effect in the solid phase. When comparing Figure 3 and Figure 4 it is visible that for dielectric constant data for pure 12CB and 12CB + C_60_ composite overlaps. This fact indicates that the addition of C_60_ nanoparticles influences only constant terms denoted as “$\varepsilon^{*}$“ in Equation (1). Notably, the addition of only 0.1% of C_60_ nanoparticles decreases the dielectric constant by ~20% in the isotropic liquid and ~6% in the Sm*A* mesophase.

Regarding dynamics, in the isotropic liquid phase, the single relaxation process (*alpha*, primary) dominates (Figure 5). In the Sm*A* phase, there is a clear manifestation of two relaxation modes, presented in Figure 6. The comparison of Figure 5 and Figure 6 can suggest that the δ-mode relaxation process can be considered as the successor of the *alpha* relaxation. The visual test of results presented in Figure 5 indicates the Super-Arrhenius (SA, $E_{a}\left(T\right)$) dynamics for pure 12CB and its nanocomposite with x=0.1% fullerene.

For x=0.05% nanocomposite, the dynamics is apparently Arrhenius type (the activation energy $E_{a}\left(T\right)=E_{a}=const$), shown by the straight-line portrayal in Figure 5. For this amount of C_60_ fullerene, the dynamics are also notably faster. For pure 12 CB and x=0.1%, the dynamics are apparently of the SA type (the bent line in Figure 5). As shown in Figure 6, also in the Sm*A* phase, the dynamics for x=0% and x=0.1% are related to almost the same time scales, whereas x=0.05% is visibly faster. The “visual analysis” indicates the basic Arrhenius-type evolution in the smectic A phase. It is also a considerably faster tumbling mode of relaxation. However, for this case, the addition of fullerenes strongly slows down the relaxation process.

Figure 7 shows the results of the fragility-focused analysis (Equations (8) and (9)) of experimental data for the *alpha*-relaxation in the isotropic phase and δ-relaxation in the Sm*A* mesophase. The distortion-sensitive tests are able to reveal the subtle distinction between the non-Arrhenius and the basic Arrhenius dynamics. The latter should appear as the horizontal line. It is visible that such behavior is an absent event for the “visually Arrhenius” behavior appearing in Figure 5 and Figure 6. The occurrence of “universal” behavior of the apparent fragility, given by Equation (8), is notable, although the addition of C_60_ nanoparticles introduces discrepancies in the immediate vicinity of the *I*–Sm*A* and Sm*A*–*S* phase transition. The experimental validation of Equation (8) in Figure 7 enabled the portrayal of the clearly non-Arrhenius behavior in Figure 5 via the “mixed” (“activated”, “critical”) Equation (7). Singular temperatures $T^{*}$ were determined using results presented in Figure 6, via the condition: $1/m_{P}\left(T^{*}\right)=0$, [39].

Finally, we discuss the phase behavior of mixtures using our mesoscopic modeling. We first consider how NPs affect pretransitional behavior above *T*c. Afterward, we discuss reentrant Super-Arrhenius (SA)-Arrhenius–Super-Arrhenius (SA) behavior that occurs when the concentration c of NPs increases. Above the SmA-I phase transition temperature, *T*c terms are linear and quadratic in the nematic degree of ordering dominate behavior in $\bar{f}$. These contributions are expressed as $\Delta \bar{f}=a^{\left(n\right)}\left(T-T_{n}^{*}\right){\bar{S}}^{2}-\sigma \bar{S}$, where we take into account Equations (10), and $\sigma =\varepsilon_{0}\Delta \varepsilon E^{2}+pw/r$. Minimization of $\Delta \vec{f}$ yields the degree of paranematic ordering above *T*c:(20)$$\bar{S}\approx \frac{\sigma }{a^{\left(n\right)}\left(T-T_{n}^{*}\right)}=\frac{\varepsilon_{0}\Delta \varepsilon E^{2}+pw/r}{a^{\left(n\right)}\left(T-T_{n}^{*}\right)}=\frac{\xi_{n}^{2}}{\xi_{E}^{2}}+p\frac{\xi_{n}^{2}}{rd_{e}}$$

Therefore, the para-nematic ordering is linearly proportional to *p*. Next, we discuss possible reasons for the SA–Arrhenius–SA crossover behavior on the increasing concentration of NPs. The measurements reveal some glassy features in the pure 12CB sample. In the presence of a relatively weak concentration of NPs (*x* ≈ 0.0005) glassy features become negligible, suggested by the Arrhenius-type behavior of the dominant structural relaxation time τ. However, for *x* ≈ 0.001, glass-type (SA dynamics) characteristics already reappear. A possible explanation is as follows. Note that pure Sm*A* exhibits quasi long-range order due to Landau–Peierls instability. Because of its smectic layer, fluctuations diverge logarithmically with the system size. These relatively strong thermal fluctuations and the mosaicity of the system introduce some randomness, and consequently, the observed glassy features (i.e., the VFT-type temperature behavior of τ). For a relatively weak concentration of NPs, which we refer to as the diluted regime, the presence of NPs effectively suppresses the smectic layer fluctuations. Consequently, the degree of randomness decreases, resulting in the prevailing Arrhenius-type behavior in $\tau \left(T\right)$. To demonstrate this effect, we consider key terms in free energy expression given by Equations (10). We consider terms related to the nematic director, which we represent by the angle $\theta =arc\mathrm{Cos}(\vec{\vec{n}}\circ \vec{v})$. In the ideal Sm*A* phase, it holds that θ=0. Note that the smectic layer bending term tends to align with the smectic layer normal along $\vec{n}$. Therefore, fluctuations in $\vec{n}$ are, for a sufficiently strong constant C, strongly coupled with fluctuations in ϕ. For relatively weak fluctuations in $\vec{n}$, it holds that
(21)$$\Delta f=\frac{k}{2}{\left|\nabla \theta \right|}^{2}+Cq_{0}^{2}\eta^{2}\theta^{2}$$where we took into account only the free energy terms, which depend on $\vec{n}$ up to the second-order expansion in θ. We expand θ using the Fourier decomposition $\theta =\sum \theta_{q}^{}e^{i\vec{q}\circ \vec{r}}$. We introduce it into the free energy term $\Delta F={\int \Delta fd}^{3}\vec{r}$, where we neglect fluctuations in order parameter amplitudes *S* and η. It follows that $\Delta F=VCq_{0}^{2}\eta^{2}{\sum \left|\theta_{q}\right|}^{2}\left(\frac{q^{2}\lambda^{2}}{2}+1\right)$. Taking into account the equipartition theorem, we obtain the equation for the amplitude of fluctuations $\left|\theta_{q}\right|=\frac{\theta_{0}}{\sqrt{1+q^{2}\lambda^{2}/2}}$, where $\theta_{0}=\sqrt{k_{B}T/VCq_{0}^{2}\eta^{2}}$. Note that fluctuations $\vec{n}$ do not diverge in the long-wavelength limit q=0 due to condensation of smectic layers, which act on $\vec{n}$ as an external field (i.e., nematic Goldstone fluctuations become massive in the SmA phase, related to the Higgs-type mechanism). The average degree of nematic fluctuations is represented by
(22)$$\langle \theta^{2}\rangle =\frac{1}{v}\int \theta^{2}d^{3}\vec{r}={\sum \left|\theta_{q}\right|}^{2}$$where $\langle \ldots \ldots \rangle$ stands for the ensemble averaging. The sum is realized over all possible wave vectors $\overset{↔}{q}$, i.e., in the amplitude interval $\left[q_{\min},q_{\max}\right]$. Here, $q_{\min}\approx 2\pi /R\approx 0$, where *R* stands for the characteristic linear sample size, and $q_{\max}\approx 2\pi /d_{0}$. If NPs are present, they reduce the fluctuation spectrum. Consequently, there are fewer contributions in the summation (over positive), and thus, the value $\langle \theta^{2}\rangle$ is reduced. In a rough approximation, we set that, in the presence of NPs, the lower integration limit is increased to $q_{main}\approx {2\pi /l}_{NP}$. Consequently, in the diluted regime, one expects Arrhenius-type dynamics. With increasing c we enter the distorted regime, where NPs are strong enough to globally distort the LC sample. Below we derive an estimate for the critical condition of this phenomenon. In our rough estimate, we approximate the LC structure in the diluted regime by a bulk-like uniform Sm*A* structure, which we refer to as the uniform structure. The competing distorted structure is characterized by apparent elastic distortions in the LC medium. We set the parameters so that they appear to accommodate local ordering tendencies at LC–NP interfaces. In the following, we estimate the average free energy density penalties of uniform structure ($\overset{↼}{f}=f_{US}$) and distorted structure ($\overset{↼}{f}=f_{DS}$). The critical (or crossover) condition, where apparent structural changes in LC ordering appear, are inferred from the condition ${\overset{↼}{f}}_{US}=f_{DS}$. The key free energy density contributions in this competition are the smectic elastic $f_{e}^{\left(s\right)}$ and NP–LC interface term *fi*, see Equations (10). In the uniform structure, if holds that $\langle f_{e}^{\left(s\right)}\rangle =0$, However, the nematic director, in general, does not match easy directions at LC–NP interfaces. It follows that $f_{i}=w\bar{S}/2\left(3{\left(\vec{n}.\vec{v}\right)}^{2}-1\right)$, yielding ${\overset{↼}{f}}_{US}\approx 0$. On the other hand, we set the parameters such that in the distorted structure, the homeotropic anchoring is obeyed at NP–LC interfaces, thus ${\bar{f}}_{i}=-w\vec{S}$. Consequently, elastic distortions must be introduced in the LC medium. We assume that the characteristic distortion length scale of elastic distortions is equal to the average separation *l_NP_* between NPs. It follows that ${\bar{f}}_{e}^{\left(s\right)}=C{\bar{\eta }}^{2}q_{0}^{2}\frac{d_{0}^{2}}{l_{NP}^{2}}$, and $f_{DS}=C{\bar{\eta }}^{2}q_{0}^{2}\frac{d_{0}^{2}}{l_{NP}^{2}}-wp\bar{S}$. The condition $f_{DS}=f_{US}$ could be rewritten into the expression
(23)$$\frac{d_{e}d_{0}^{2}}{r\lambda^{2}}={\left(\frac{4\pi }{3}\right)}^{2/3}p^{1/3}$$

For *p* = 0.001, it follows that $d_{0}d_{e}^{2}/\lambda_{0}^{2}r\approx 0.3$, which is a sensible value. Therefore, for a high enough concentration, a glass-like structure is expected, which is fingerprinted in the Super-Arrhenius dynamics.

## 4. Conclusions

Generally, studies of the temperature evolution of the dielectric constant enable direct insight into the orientational, uniaxial, ordering that comprise the key symmetry features of both nematogenic and smectogenic rod-like liquid crystalline compounds. This report, in agreement with earlier studies of the author, clearly shows that the parameterization of the dielectric constant in the smectic A and isotropic phase is totally dominated by the pretransitional effect associated with the *I*–Sm*A* weakly discontinuous phase transition. For the isotropic phase, this can be clearly related to the increasing amount of order, in an antiparallel manner, of permanent dipole moments within “internally ordered prenematic fluctuation”. This leads to a decrease in $\varepsilon \left(T\to T_{ISmA}\right)$, $T>T_{ISmA}$. On the low-temperature side of the clearing temperature, there is a long-range increase in $\varepsilon \left(T\to T_{ISmA}\right)$, $T<T_{ISmA}$. This can be explained by isotropic fluctuations—heterogeneities appearing in the isotropic “matrix”—with the notably larger dielectric constant compared to the surrounding constants dominated by the antiparallel arrangement. Consequently, should the smectic A phase in 12CB be considered rather as the kind of “critical homo-composite” with isotropic fluctuations—”droplets”—that are heterogeneities appearing within the basic Sm*A* structure? Such a “homo-colloidal” picture seems to take place also in the isotropic liquid phase as well as in the nematic phase.

The relatively small addition of C_60_ fullerene nanoparticles strongly influences dynamics and phase transition. For the latter, one can indicate strong changes in metrics of the discontinuities of the I–SmA transition ($\Delta T^{*},\Delta T^{**}$) and the “general” value of the dielectric constant, which is realized mainly via the shift of $\varepsilon^{*}$ and $\varepsilon^{**}$ coefficient in Equations (1) and (2). For dynamics, the ability of C_60_ dopants to facilitate the crossover from the Super-Arrhenius to the Arrhenius pattern is notable.

Increased values of *T*_ISmA_ with respect to the bulk sample reveal that for the range of studied concentrations, NPs effectively support the onset of orientational ordering. On the other hand, they effectively suppress phase transition into the crystal phase, which is fingerprinted in an apparent decrease in the crystallization phase temperature. Next, we observe an increasingly suppressed dielectric response with increasing *x* values. Note that for ferroelectric NPs [10], different behavior is observed with varying x values due to the inherent polarization of NPs in the latter case. Furthermore, with increasing *x* values, we observe intriguing behavior in dynamics in the isotropic phase. Namely, in bulk and for *x* = 0.001 we obtain SA-type dynamics, and for *x* = 0.0005, one observes an Arrhenius type behavior. In general, SA fingerprints glass-type features. Our measurements reveal some kind of reentrant behavior. With increasing *x* values, the glass-type characteristics are first suppressed (x~0.0005), and for large concentrations (x~0.001), SA behavior is recovered. Namely, NPs could efficiently suppress smectic fluctuations in the isotropic phase. NPs are expected to shrink the spectrum of nematic director field fluctuations. Note that $\vec{n}$ tends to be aligned along with the corresponding smectic normal layer, and consequently suppression of fluctuations in $\vec{n}$ indirectly also suppresses fluctuations in smectic layers. Furthermore, NPs in our study seem to be coupled with the degree of LC ordering, which is fingerprinted in an increase in $T_{ISmA}.$ Therefore, qualitative change in glassy behavior could be due to the dominance of NP-driven suppression of layer fluctuations for x≈0.0005 and the dominance of NP-driven disorder for higher concentrations.

In summary, we would like to highlight the new results presented in this article:
The evidence showing that even a tiny (very small concentration) addition of fullerenes can strongly change the dielectric constant (coupled to the arrangement of dipole moments) and the maximum of the primary loss curve (coupled to the energy associated with the given relaxation process).The temperature evolution of the above properties is related to the isotropic–Smectic A transition, in the liquid phase up to $T_{ISmA}+65K$ and in the whole smectic phase.The addition of fullerenes does not change the functional forms of pretransitional anomalies, particularly the value of the “critical” exponent.There are hallmarks of the pretransitional effect for the Sm*A*–Solid transition, enhanced by the addition of fullerenes.The increase in the number of fullerenes shifts dynamics from the clear SA (“glassy”) pattern to the (almost) Arrhenius pattern, and finally back to SA scaling again.The Landau–de Gennes phenomenological model is the base for portraying phase transition impact in liquid crystals, mainly related to thermodynamic and static properties. This report shows that it can be extended to describe composites of smectogenic LC and nanoparticles (fullerenes) and obtain the SA-A–SA crossover in dynamics when changing the concentration of fullerenes.

Finally, we would also like to attract attention to the new way of analysis based on the apparent fragility index and the new “mixed” equation linking the activated (exponential) and critical-like terms—issues which have been not addressed so far.

## Figures and Tables

**Figure 1 nanomaterials-10-02343-f001:**
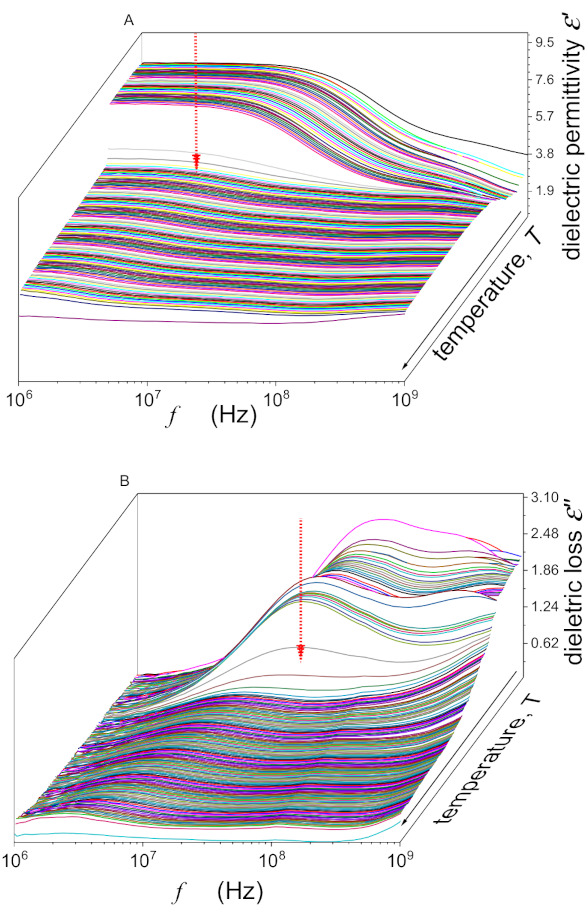
The general view of temperature and frequency evolutions of dielectric spectra for the real ($\varepsilon^{\prime }$) and imaginary ($\varepsilon^{\prime \prime }$) components of dielectric permittivity in pure 12CB. Curves are related to frequency scans for subsequent temperatures. Note the appearance of loss curves for $\varepsilon^{\prime \prime }\left(f\right)$ and the static (horizontal) domain for $\varepsilon^{\prime }\left(f\right)$. The latter is related to the dielectric constant. Red arrows indicate Isotropic–Smectic A “clearing” temperature.

**Figure 2 nanomaterials-10-02343-f002:**
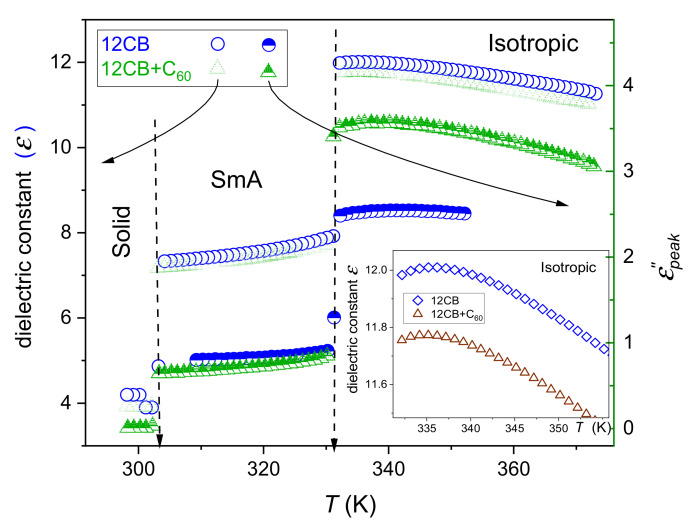
Temperature evolutions of the dielectric constant $\varepsilon =\varepsilon^{\prime }\left(T,f=10\;\mathrm{kHz}\right)$ and primary loss curves’ maxima $\varepsilon_{peak}^{\prime \prime }\left(f,T\right)$, 100 Hz<f<1 GHz for 12CB and 12CB + C_60_ fullerene (0.1% mass fraction) nanocomposite. The inset shows the behavior of the dielectric constant in the isotropic liquid phase, for the immediate vicinity of the clearing temperature.

**Figure 3 nanomaterials-10-02343-f003:**
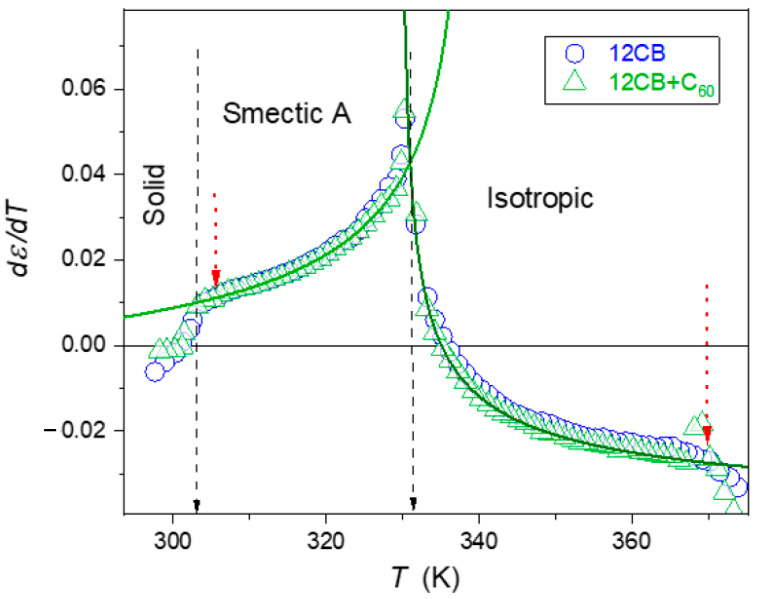
The temperature *behavior* of the derivative of the dielectric constant in subsequent phases of liquid crystalline 12CB and the composite 12CB + C_60_ (0.1%) fullerene. The plot was obtained based on experimental data given in Figure 2. Solid curves are related to Equations (16) and (18). Solid arrows indicate phase transition temperatures. Short, dashed arrows indicate the onset of distortions remote from the *I–N* transition and near the Sm*A*–*S* transition.

**Figure 4 nanomaterials-10-02343-f004:**
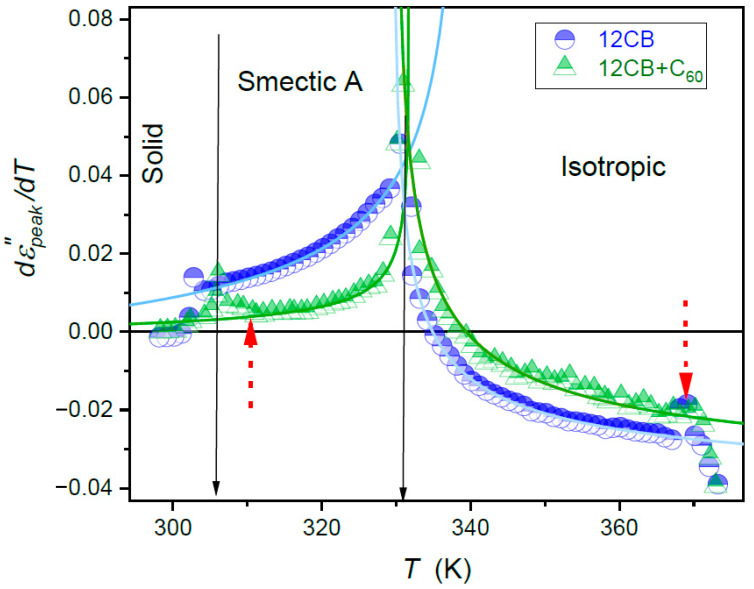
The temperature behavior of the derivative of the maximum of the primary dielectric loss curve in subsequent phases of liquid crystalline 12CB and the composite 12CB + C_60_ (0.1%) fullerene. The plot has been obtained based on experimental data given in Figure 2. Solid curves are related to Equations (17) and (19). Solid arrows indicate phase transition temperatures. Short, dashed arrows indicate the onset of distortions remote from the *I–N* transition and near Sm*A*–*S* transition.

**Figure 5 nanomaterials-10-02343-f005:**
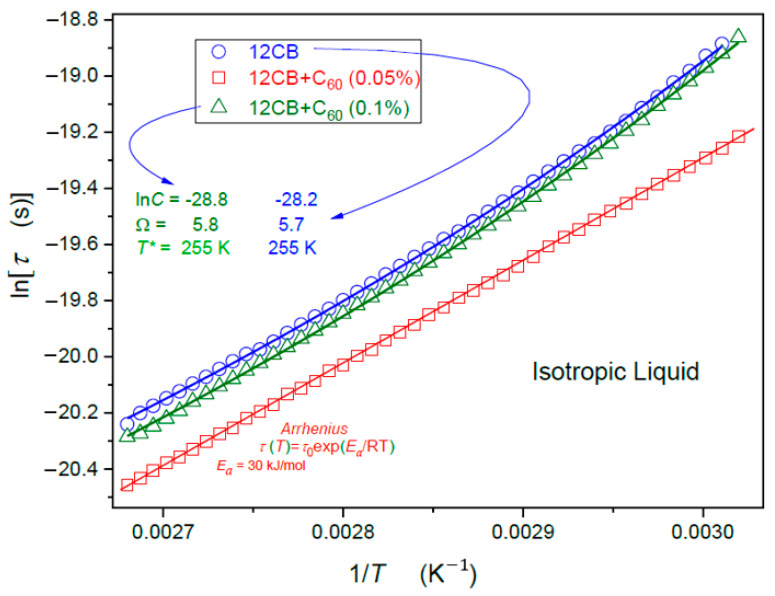
The evolution of primary relaxation times in the isotropic liquid phase of 12CB and its nanocolloids with C_60_. For 12 CB with *x* = 0 and *x* = 0.1%, a fair portrayal by the SA Equation (1) is obtained (solid curves). Fitted parameters are given. For 12CB + *x* = 0.05% of C_60_ composites, experimental data follows the Arrhenius relation with the activation *Ea* = 30 KJ/mol.

**Figure 6 nanomaterials-10-02343-f006:**
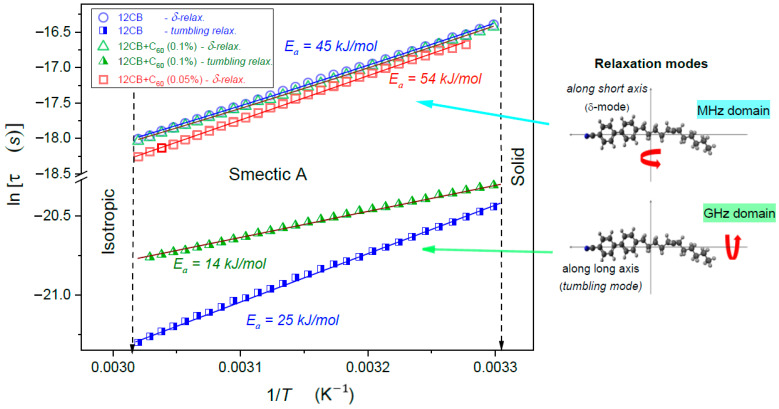
The relaxation map in Sm*A* mesophase of 12CB and its nano colloids with C_60_ fullerenes. The “long-time” relaxation process is portrayed by the SA Equation (1). The short duration of “tumbling” relaxation follows the basic Arrhenius pattern with activation energies given in the Figure.

**Figure 7 nanomaterials-10-02343-f007:**
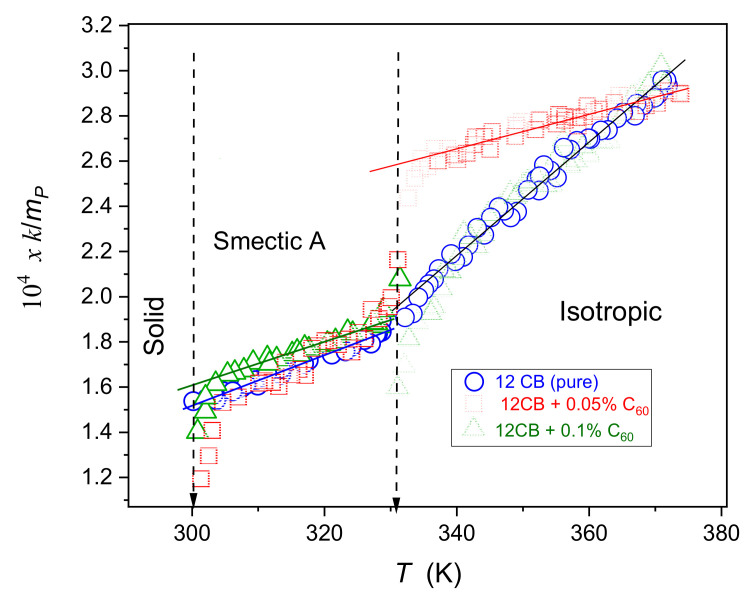
Temperature evolutions of the apparent fragility in the 12CB and its nanocolloids with C_60_ fullerenes.

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
