# Peer review of "Dynamics and Pretransitional Effects in C60 Fullerene Nanoparticles and Liquid Crystalline Dodecylcyanobiphenyl (12CB) Hybrid System"

_nanomaterials, 2020, doi:10.3390/nano10122343_

Round 1

Reviewer 1 Report

The aim of paper is to identify the effects of doping the rod-like liquid crystal dodecylcyanobiphenyl (12CB) with buckminsterfullerene (C60). The main contributions are related to the impact of C60 on phase transitions of 12CB.

General comments

* The article needs to be rewritten and restructured. Please follow the suggestions below:

The abstract should be reformulated: more precise and more structured.

The Introduction should be rewritten in a more clearly version. The State-of-Art must be focused on LCs-NPs composites. Highlight targets and applications. Avoid direct references to the “authors of this report”. Also avoid using the word metamaterial in association with a (nano)composite.

The chapter “2. Material and Methods” should be restructured.

Insert the sub-chapter 2.1 “Samples preparation” and fills it in with all necessary information about materials used, design of samples and experimental setup (a picture is usefully) and experimental procedure of samples preparation.

In the next sub-chapter: 2.2 introduce only information about experimental measurement procedures, including exact types of apparatus.

In the last sub-chapter “The analysis of experimental data” insert only information about procedures of experimental data analysis, including the software used and the computer architecture (if it is relevant). Here is the place for a very brief description of the models used in the analysis, also for unpublished updated models.

In the chapter “3. Results”, put only the experimental result and avoid discussing the results. Here is the place of the detailed description of unpublished developed model.

Then insert chapter "4. Discussion" with all the discussion about the experimental results and the comparison with the models.

In the chapter "Conclusions", collect all the conclusions.

* Other general comments and recommendations.

Define each symbol used in the equations.

How good fit the model curves the experimental data? Give an error analysis.

Check the English avoid expressions that seen colloquial.

There are to too many autociting references. Try to limit autocitation to 25%.

Specific comments

Keyword: dynamics is too general. Find another.

In chapter 2

Row 71: insert a reference for permanent dipole moment.

Also include references for each physical property value.

In chapter 3

Insert a plot with the typical temperature dependence of the complex permittivity

References

Reference [1] is a collection of articles. Specify which articles are cited, in separate references. Also check other book references.

Author Response

Reviewer 1

  • The reviewer suggested to include 4 additional references.
  • They have been added (current refs. [40-43]) + few others continuing the same reasoning strictly for fullerenes. The related comment is given in the Conclusions section. It is also stressed that none of these reports address exactly issues presented in the given Report, namely: (i) the characterization of the pretransitional effect (ii) the advanced characterization of the evolution of relaxation times.
  • The issues ‘what happens when increasing the concentration ‘ is addresses in the experimental section. It is stated that even raising the concentration of C60 to 10% did not create problems for SmA phase.
  • The issue of molecular systems vs. (nano-colloidal) systems, in fact is one of the topics of the report, which shows that essentially the description of pretransitional effects and dynamics is similar but nanoparticles (C60 in the given case) open possibilities of a ‘smooth’ moderating of key parameters.

Reviewer 2 Report

The Authors present an experimental investigation concerning the properties of 21CB (a typical smectogenic liquid crystal) doped with C60 fullerene.

They have studied the dielectric behaviour and pre-transitional effects. The Authors find that already with a relatively small amount of C60 added to the LC compound, a 20% change in the dielectric constant was observed. Moreover, they are able to assess the impact of pre-transitional effects near the isotropic-to-smectic and within the smectic phase itself.

The work is very nice and well conducted and definitely adds to the field.

The subject is quite interesting since 12CB and C60 are of very different shape but comparable size and excluded volume effects should dominate the behaviour of the mixture. The Authors should make a comparison with the effects predicted by MD simulations of mixtures of simple rod-like and spherical particles, e.g. J. Chem. Phys. 120, 10307 (2004) (neutral particles), Soft Matter 2017, 13, 5207-5213; Phys. Chem. Chem. Phys. 2019, 21, 20327-20337 (both neutral as well as charged particles), although the concentrations of nanoparticles in the experiments run by the Authors is much lower.

Did the Authors try to increase the concentration of C60 fullerene in the LC phase? At which concentration is a phase separation observed.

Moreover, experimental data on colloidal mixtures of rods and spheres have been reported, see for example Soft Matter, 2016, 12, 9238. How the results obtained on the molecular mixtures used by the Authors compare with the colloidal systems?

Author Response

Reviewer 2

  • The new organization ‘chapters – subchapters’ was suggested
  • It has been done, which was associated with some significant rearrangement of the paper, to preserve the conclusive coherence.
  • The error analysis is often a ;soft’ problem of many papers. In fact, any non-linear and multiparameter.  Fitting of experimental data is often associated with ‘too large – to show’ .fitting error. On the other way, the residual analysis is hardly conclusive because often few equations yield a comparable quality

 In this paper this issue has been overcome via two – innovative – ways. Both for pretransitional effects and for dynamics the distortions-sensitive analysis is used what enable unequivocal determining of parameters and shows the real error, reducing also the number of fitted parameters.  See lines 193-201.

  • Defining of all symbols, English language quality has been carefully tested.
  • Recalling to the authors’, reference has been reduced. But one should stress that the given report addresses problems (i) the characterization of the pretransitional effect for dielectric properties (ii) the fragility – supported description of dynamics. For both issues, one of the authors (A. Dr0zd-Rzoska) is the leader in introducing and developing topics, and then the notable recalling of references is natural. Notwithstanding, we did the best to approach suggestions regarding the number of references, reducing them. Comments are included in the experimental Section.
  • According to the suggestion ref. [1] is deeper specified
  • ‘Insert a plot with the typical temperature dependence of the complex permittivity’
  • Such a plot can have only an illustrative meaning but can be o=important fr a non-experienced listener. Please see the new 1 based on experimental data used for the given research and presents 3D pictures for the real and imaginary parts of dielectric permittivity.

Hence, the following Figures had to be renumbered, and now there is 7 Figures.

Reviewer 3 Report

The presented manuscript “Dynamics and pretransitional effects in C60 fullerene nanoparticles and liquid crystalline dodecylcyanobiphenyl (12CB) hybrid system” written by S.J. Rzoska et al., which was submitted for publication in Nanomaterials, reports on new liquid crystalline nanocomposite.

The presented manuscript deals with classical liquid crystal 12CB admixed with small amount of fullerene. I found its subject as attractive and topical. Nevertheless, I have objections mostly concerning the data presentation and lack of suitable Summary or Conclusions.

In the Introduction, I cannot find information about previous results obtained for fullerene nanocomposites.

I do not like that authors are presenting lot of technical details. On contrary, the explanation and interpretation of these data do not satisfy me, as it is not straightforward and/or clear.

Line 101: I know that 1-1/2=1/2, but I do not know why alpha=1/2, as it is not clear in the context later. I do not know if coefficient phi is the same for all calculations.

I missed discussion of the fact that the system with concentration 0.1% is closer to the pure compound than the concentration 0.05%. Is it possible that agglomeration took place?

I missed a real summary of results. In Abstract, authors claimed that they report a strong impact of fullerene. I am not convinced that authors really did, I do not see this fact properly presented and demonstrated.

I completely missed any conclusions. To finish with the equation (20) and the statement that some particular value is reasonable, it does not satisfy any scientific merit.

No Summary, no Conclusions, it is not acceptable.

I recommend major revision of the manuscript.

Author Response

Reviewer 3

  1. Reviewer: In the Introduction, I cannot find information about previous results obtained for fullerene nanocomposites.

The response: note the current recalling of references in Introduction and Conclusions. The latter also includes 10 new references. They are related to fullerenes, although nome of them is related to the in-deep analysis of pretransitional effect and the glassy , complex dynamics. In our opinion, this is the first such paper

  1. Reviewer: I do not like that authors are presenting lot of technical details. On contrary, the explanation and interpretation of these data do not satisfy me, as it is not straightforward and/or clear.

The response: please see the current organization and presentation of results , the new Fig. 1 and the discussion in lines 190-202.  All these are associated with clear recalling to earlier reports.

  1. Reviewer Line 101: I know that 1-1/2=1/2, but I do not know why alpha=1/2, as it is not clear in the context later. I do not know if coefficient phi is the same for all calculations

The response: it is difficult to repeat issues explained 10x or more in earier reports, from the classical paper by Thoen et al. (1981).  There are clear references in the text to these papers, in which the link between the exponent alpha for the specific heat and the anomaly of dielectric constant is explained

  1. Reviewer: I missed the discussion of the fact that the system with concentration 0.1% is closer to the pure compound than the concentration of 0.05%. Is it possible that agglomeration took place?

The response: this issue – still challenging – is now recalled in Conclusion (also due to indications of Reviewer 1) that nanoparticles can agglomerate in a specific way leading to orientation/distortions of the LC matrix. On the other hand, this issue is in deep commented by the further (new ) extension of the theoretical model end of the Results and Discussion Section.

  1. Reviewer: I missed a real summary of results. In Abstract, the authors claimed that they report a strong impact of fullerene. I am not convinced that the authors really did, I do not see this fact properly presented and demonstrated.

The response: I think that the Reviewer should ones more pay attention to Figures showing  the impact of fullerenes on dielectric constant and particularly on the maximum of loss curve, which is related to the energy coupled to the given relaxation process/mode.  The next issue in the clear crossover  Super-Arrhenius – Arrhenius – SuperArrhenius dynamics when changing the amount of C60.

Finally, the strong slowing down or speeding-up of alpha- , delta and tumbling relaxation when changing concentrations.

These facts – very clearly shown – are in clear and strong disagreement with the ‘decisive statement’ of the Reviewer.

  1. Reviewer: ‘I completely missed any conclusions. To finish with the equation (20) and the statement that some particular value is reasonable, it does not satisfy any scientific merit No Summary, no Conclusions, it is not acceptable

The response: Please see the current Results & Discussion section as well as the Conclusion. There is a very in deep extension which clearly responds to this point.

Round 2

Reviewer 1 Report

The authors did not answer all of Reviewer 1's questions. Some answers are only partial answers.
In the Conclusions, the authors should clearly highlight the new results presented in this article.

Author Response

Reviewer 1

Comment of Reviewer:  ‘The authors did not answer all of Reviewer 1's questions.  Some answers are only partial answers. In the Conclusions, the authors should clearly highlight the new results presented in this article.’

Response: Following the 1st opinion we can guess that ‘1st question’ can be related to the Abstract and the Introduction. They have been strongly rearranged and cleaned. There are clear distinctions between ‘other’ nanoparticles and fullerenes. Applications are clearly recalled. There is a clear split between theory and experiments. Finally, the novelty the reader should expect in the paper is clearly stated:  the distortions-sensitive insight into pretransitional effects and the (complex) dynamics.

At the end of the Conclusions section note line 487: ‘In summary, we would like to highlight the new results presented in this article:’  -  they are given (in points) in lines 488 – 504.

The above changes required renumbering of references, with removing a ‘double reference’ .  The language has been in deep tested and corrected.

Reviewer 3 Report

The presented manuscript “Dynamics and pretransitional effects in C60 fullerene nanoparticles and liquid crystalline dodecylcyanobiphenyl (12CB) hybrid system” written by S.J. Rzoska et al., submitted for publication in Nanomaterials, was revised now.

I have only two remarks:

Line 117: I do not know why authors insisted in writing 1-1/2=1/2. I surely prefer 1-alpha=1/2 when taken alpha=1/2 from xxx measurements for example. It looks really funny, as everybody knows that 1-1/2=1/2.

Line 470: I do not understand meaning of the sentence: “We believe that the reason behind these two contradicting competing mechanisms introduced by NPs.” Does it mean “is introduced”?

In any case, the form of manuscript corresponds better to the standard of scientific paper now.

I am satisfied.

I can recommend the manuscript for publication.

Author Response

Reviewer 3

There were only 2 comments:

  1. Comment of Reviewer: I do not know why authors insisted in writing 1-1/2=1/2. I surely prefer 1-alpha=1/2 when taken alpha=1/2 from xxx measurements for example. It looks really funny, as everybody knows that 1-1/2=1/2.

Response: This has been corrected exactly following the advice. See the current line 141

  1. Comment of Reviewer: ‘I do not understand meaning of the sentence: “We believe that the reason behind these two contradicting competing mechanisms introduced by NPs.” Does it mean “is introduced”?

Response:  This sentence has been removed. It did not introduce any value for reasoning in the report.

There are also quite extensive corrections (introduction and conclusions) associated with the comments of Reviewer 1. But they did not change responses to suggestions of  Reviewer 1  in the First Opinion.  No new interpretations, no new comments. Only rearrangements, which have led also to ‘renumbering of references.  The language has been also finally corrected.